# A New Unified Theory of Trigger Point Formation: Failure of Pre- and Post-Synaptic Feedback Control Mechanisms

**DOI:** 10.3390/ijms24098142

**Published:** 2023-05-02

**Authors:** Robert D. Gerwin

**Affiliations:** Department of Neurology, The Johns Hopkins School of Medicine, Baltimore, MD 21287, USA; bbgerwin@gmail.com

**Keywords:** feedback mechanism, myofascial pain syndrome, sympathetic nervous system, trigger points, ion channelopathy, neuromuscular junction, acetylcholine, excitation–contraction coupling

## Abstract

The origin of the myofascial trigger point (TrP), an anomalous locus in muscle, has never been well-described. A new trigger point hypothesis (the new hypothesis) presented here addresses this lack. The new hypothesis is based on the concept that existing myoprotective feedback mechanisms that respond to muscle overactivity, low levels of adenosine triphosphate, (ATP) or a low pH, fail to protect muscle in certain circumstances, such as intense muscle activity, resulting in an abnormal accumulation of intracellular Ca^2+^, persistent actin-myosin cross bridging, and then activation of the nociceptive system, resulting in the formation of a trigger point. The relevant protective feedback mechanisms include pre- and postsynaptic sympathetic nervous system modulation, modulators of acetylcholine release at the neuromuscular junction, and mutations/variants or post-translational functional alterations in either of two ion channelopathies, the ryanodine receptor and the potassium-ATP ion channel, both of which exist in multiple mutation states that up- or downregulate ion channel function. The concepts that are central to the origin of at least some TrPs are the failure of protective feedback mechanisms and/or of certain ion channelopathies that are new concepts in relation to myofascial trigger points.

## 1. Introduction

The myofascial trigger point (TrP), considered to be the underlying cause of myofascial pain syndromes, was described by Travell and Rinzler in 1946 [1]. It has been increasingly studied since, despite a controversy over its existence [2]. Clinical presentations and management regimens are well-represented in the literature, but there has been little or no discussion of its origin. This paper proposes a new hypothesis of trigger point formation, one that has not appeared in the literature previously. It is acknowledged that there may be multiple causes for TrP formation [3], all leading to a same result, the TrP, and that the mechanisms proposed here may apply only to a subset of TrPs. 

The specific mechanisms by which the TrP develops remains unknown despite descriptions of mechanical and physiologic stresses that are predisposed to and maintain the TrP, such as the perpetuating factors identified by Travell and Simons [4]. Simons proposed an integrated TrP hypothesis that implicated an energy crisis as a major factor causing TrPs. His hypothesis, based on (1) the absence of motor action potentials, (2) the activation of TrPs by muscle overload, (3) nociceptor activation, and (4) the therapeutic effect of stretching, postulated an overactive neuromuscular junction (NMJ) that releases excessive acetylcholine (ACh) in response to muscle overload ([4] pp. 68–79). He further postulated that the capillary compression from TrPs leads to ischemia and an inability to replenish adenosine triphosphate (ATP) that prevents the reuptake of Ca^2+^ from the muscle cytosol, resulting in persistent muscle sarcomere contractions. The integrated hypothesis of Simon, useful as it was, did not in fact address the actual mechanisms by which the TrP is formed. It did not account for the many regulatory mechanisms that protect muscle from injury. Even the updated versions of Simons’ integrated hypothesis left the actual origin of the TrP as a ‘black box’ mystery [5,6]. The new hypothesis proposed here is based on physiologic mechanisms that could play a role in the genesis of the TrP in at least a subset of cases. The new hypothesis is derived from a review of the current literature. It proposes that there is a failure of protective regulatory mechanisms, many of them feedback mechanisms, that prevent excessive muscle activity or that prevent a potentially injurious accumulation of Ca^2+^ within the muscle cytosol and offers specific examples to support the hypothesis. The new hypothesis relates these mechanisms to the development of the TrP in a way that has not been described in the literature before. It is offered as a means of stimulating research into the physiology of TrP formation and has implications for the nature of unusual fatigue and weakness in muscles with TrPs, as well as treatment implications. 

Myofascial TrPs were described about 75 years ago in a paper on non-cardiac chest pains [1], though there were descriptions of similar phenomena prior to that time. Subsequent publications described referred to pain patterns and the effect of manual or invasive therapy (dry needling or TrP injection). Early histopathological reports of muscle hardening or myogelosis in TrP regions found focal areas of swelling in muscle [7,8,9,10]. Electrophysiologic studies of the TrP began to be published in the 1990s [11,12,13]. A microanalytic technique that explored the extracellular biochemical milieu of the TrP opened a new means of investigation [14]. TrP imaging, long an elusive goal, is accomplished by magnetic resonance elastography [15] and high-definition ultrasound [16]. Histopathological studies of TrPs in humans remains sparse but suggests that segmental sarcomere contraction occurs at the TrP or in adjacent taut bands (TB) [17,18].

## 2. Trigger Point Physiology

This section summarizes the present body of evidence for TrP physiology, imaging, and structure, and suggests some implications of the findings (Table 1).

### 2.1. Electrophysiology

Normal resting muscle is relatively electrically quiet. Miniature endplate potentials (MEPPs) occur at a frequency of 1–6 per second and endplate spikes occur in resting muscle [19,20]. In contrast to resting muscle, the electromyogram (EMG) of the TrP shows a persistent, low amplitude (5–50 µV), high frequency activity that looks like high frequency MEPPs punctuated by intermittent, higher amplitude (100–600 µV), that are initially negative, biphasic, endplate spikes [13]. Resting TrP EMG activity, termed endplate noise (EPN), may be as much as two to three orders of magnitude faster than normal MEPP frequency. EPN indicates that there is an excess of ACh molecules at the NMJ endplate zone in TrPs compared to normal resting muscle, suggesting that there might be a failure of the feedback mechanisms that regulate the release of ACh from the MNT. Alpha-adrenergic inhibitors and botulinum toxin both reduce EPN activity [12,13,21], indicating that EPN is the result of ACh released from presynaptic vesicles. 

### 2.2. Sympathetic Nerve Inhibition of TrP EPN

The ⍺-adrenergic inhibitor phentolamine reduces the average integrated signal of EPN by about 60% [13], but the specific mechanism by which this occurs has not been previously addressed. Both the ⍺- and β-sympathetic nervous systems play a role in modulating muscle contraction, as will be discussed subsequently.

### 2.3. Biochemical Pathophysiology

The TrP extracellular milieu is acidic (pH in the range of 4–5, below the normal range of 7.35–7.45) and has elevated levels of cytokines and neurotransmitters, such as IL-6, bradykinin, substance P, and calcitonin-gene-related-peptide (CGRP) compared to non-trigger point regions [14]. The acidic pH suggests that the TrP region is hypoxic and ischemic [20]. An acid extracellular milieu can inhibit acetylcholinesterase (AChE), and therefore can contribute to an increase in the concentration of ACh molecules at the motor endplate. CGRP can increase the quantal size of ACh released from the motor nerve terminal [22] and can upregulate nicotinic ACh receptors (nAChRs) at the motor endplate region, thereby expanding the AChR zone [23]. Thus, CGRP co-released with ACh from the MNT has the effect of increasing the number of ACh molecules at the motor end plate [24]. The CGRP effect at the NMJ is delayed, though not immediately, but it could be an upstream initiating event contributing to the development of the TrP.

Neurotransmitters and cytokines present in high concentration in the extracellular TrP milieu, such as substance P and CGRP may also produce neurogenic edema. The hypoechoic appearance of the TB on high-definition ultrasound is consistent with neurogenic edema, although there are other explanations for the nodular swelling at the TrP, including the recent finding of glycosaminoglycans surrounding contraction knots in an experimental TrP paradigm [25,26,27,28,29]. These possible causes of the stiffened taut band differ from the long-held idea that the taut band is the result of multiple short loci of contracted sarcomeres alternating with long zones of stretched sarcomeres.

### 2.4. Histopathological Evidence

Segmental sarcomere contraction was found retrospectively in one canine skeletal muscle specimen obtained by open biopsy from a taut band [18], and has been reported in one study of human trapezius muscle obtained by needle biopsy of a TrP region [17] performed for a study of other aspects of muscle morphology. The term ‘contraction knots’ has been used both to describe the regions of segmental sarcomere contraction and the palpable nodular hardness found within the TB ([4] pp. 67–69).

Segmental sarcomere contraction has been reported in two different animal models designed to replicate the TrP phenomenon [27,28,29]. One model utilized blunt trauma to the muscle followed by intensive exercise. The other used AChE inhibitors to induce segmental sarcomere shortening and other TrP phenomena. In addition, muscle fiber super-contraction associated with large intracellular increases in unstimulated Ca^2+^ was found in K_ATP_-deficient mouse flexor digitorum brevis single muscle fibers that were exercised to fatigue [30].

Sarcomere hypercontraction can be seen as an artifact in percutaneous muscle biopsy specimens, but this artifact does not occur when the specimens are treated with osmolarity-corrected glutaraldehyde [31], as was carried out in the study reporting on segmental sarcomere contraction in humans [17]. Anesthetics, such as lidocaine and bupivacaine can result in muscle degeneration and hypercontraction. Open biopsies are generally length-fixed and do not show segmental hypercontraction. The segmental hypercontraction in the human biopsy material cited above [17] was seen on material obtained by large bore needle biopsy, but was present both in light microscopic sections and on electron microscopy, the latter was seen in tissue fixed with osmolarity-controlled glutaraldehyde. Segmental sarcomere hypercontraction was also found in a skeletal muscle subjected to eccentrically challenged, unloaded rat adductor longus muscle [32], muscle subjected to repetitive eccentric contractions [33], and in muscle subjected to contraction and tension loading [34,35], studied both by light and electron microscopy.

### 2.5. Ultrasound Imaging of Trigger Points

Trigger points and TBs appear as hypoechoic regions on ultrasound, with retrograde blood flow seen in the TrP region itself, consistent with ischemia and consequent hypoxia [16].

**Table 1 ijms-24-08142-t001:** Objective laboratory features and subjective clinical signs of the trigger point upon which the new hypothesis is based. Specificity and sensitivity levels under physical examination are estimates, not from studies.

Feature	Description	References	Level of Confidence
Electrophysiology	High frequency, low voltage endplate noise, attenuated by ⍺-adrenergic inhibitors and by botulinum toxin. Eighteen rabbits, 10 received one dose of botulinum toxinTx, 10 received multiple doses of botulinum toxin. Ten were controls. Endplate noise was diminished in all rabbits that received BTx, but not in the control rabbits [21]. Spontaneous electrical activity was found in TrPs of 29 tension headache patients, 25 fibromyalgia patients, but in none of the eight controls [11]. Phentolamine IV injection vs control injection of saline in nine rabbits. Twenty-five active trigger spot loci were sampled. Mean average integrated signal of spontaneous electrical activity was reduced from 9.89 µV to 9.92 µV (*p* < 0.05) [13].	[11,13,21]	High
Histopathology	Segmental sarcomere contraction. 1.32 female office workers, 15 myalgic, and 15 no pain. Taut bands were found in all subjects. Sarcomere compression in five non-myaglic and two myalgic subjects on limited tissue saved from a prior study [17]. 1. Canine taut band study, 10 animals, one example identified retrospectively [20]. 2.	[17,20,23,29]	Probable, not proven
Microanalytic biochemistry	Acidic (low pH); elevated levels of certain neurotransmitters and cytokines. Study of humans with neck pain. Three controls (three latent TrPs, 3 active TrPs, no neck pain, and no TrP). Significantly elevated levels of the following in the active TrP neck pain group (*p* < 0.01): protons, BDKN, CGRP, Subs P, TNF-alpha, IL-1 beta, 5-HT, NE.	[14]	Highly likely; needs confirmation from a second laboratory
Ultrasound imaging	Nine subjects (seven women), 13 active TrP sites and nine latent TrPs sites. Fourteen normal in trapezius muscles; findings: focal, hypoechoic regions on 2D US and focal regions of reduced vibration amplitude on VSE indicating a localized, stiff nodule.	[16]	High
Magnetic Resonance Elastography	Proof of concept pilot trial on two female subjects showed taut bands that are detectable and quantifiable with MRE imaging. The findings in the subjects suggest that the stiffness of the taut bands (9.0+/−0.9 KPa) may be 50% greater than that of the surrounding muscle tissue.	[15]	High
Physical examination	Taut band, nodular region of tenderness, reproduction of usual pain; high specificity because a tender nodule on a taut band defines a trigger point. The outcome of the physical examination of trigger points remains controversial.	[4]	Moderate for diagnostic purposes; high specificity, moderate sensitivity
Physical examination	Non-wasting weakness of muscle rapidly reversed after trigger point inactivation, highly specific because improvement after release of a trigger point defines a trigger point effect. There are no studies evaluating this response.	[4]	N/A; moderate sensitivity, highly specific
History	Onset is often preceded by acute or repetitive muscle overuse.	[4,6]	n/a

Abbreviations: TrP trigger point; BDKN bradykinin; CGRP calcitonin gene-related peptide; Subs P substance P; 5-HT serotonin; NE norepinephrine; US ultrasound; VSE vibration sonoelastography.

## 3. Analysis of Elements Related to the New Trigger Point Hypothesis

### 3.1. The New Trigger Point Hypothesis

The new trigger point hypothesis proposes that in a subset of cases, the failure of protective feedback mechanisms results in a cascade of events following intensive, acute, events, such as trauma, or following chronic, repetitive, fatiguing muscle activity, resulting in the formation of TrPs. The postulated dysfunctional feedback mechanisms include those that regulate the release of spontaneous non-evoked quantal ACh at the NMJ that induce miniature endplate potentials and those that modulate the release of ionized calcium into the muscle cell, that results in the binding of actin to myosin, leading to sarcomere contraction. The new hypothesis postulates that these mechanisms fail either because they are overwhelmed to the point that feedback controls no longer respond or are insufficient, or because of up- or downregulating mutations/variants in the molecular subunits of key ion-channels that control the intracellular concentration of Ca^2+^, such as the ryanodine receptor (RyR) and the potassium-ATP (K_ATP_) ion channel, or post-translational changes that cause ‘leaky’ ion channels. Specifically, the regulatory mechanisms of interest include (1) the presynaptic ionotropic nAChR ion channels and the metabotropic muscarinic M1 and M2 receptors that modulate the release of ACh molecules into the synaptic space, and (2) the postsynaptic ionotropic potassium and sodium channels that control the influx of Ca^2+^ into the muscle cytosol. In addition, there is feedback from ACh released into the synaptic space that inhibits the presynaptic ACh release from the MNT [36] and postsynaptic activation of nAChRs that modulate ACh through a transsynaptic feedback loop. Thus, there are multiple feedback mechanisms that limit the release of ACh into the synaptic space and prevent excessively high-frequency motor unit action potentials that could overwhelm the adenosine triphosphate (ATP)-requiring calcium reuptake processes, or that could result in excessive muscle work to the point of fatigue and the development of TrPs ([4] p. 19).

Lengthening eccentric muscle contraction is an example of muscle overload that causes muscle injury [37,38,39,40,41] and/or TrPs, though the latter outcome remains unproven.

Muscle fatigue is such a dogma of TrP initiation among clinicians, that a deep understanding of the physiology of skeletal muscle fatigue would be very useful to the understanding of the origin of the TrP.

Causes of the TrP are likely multifactorial, and muscle fatigue is likely but one cause. Many factors can contribute to the vulnerability of muscle to become fatigued or to develop TrPs. One potential factor is an ineffective K_ATP_ ion channel. Mice with non-functioning K_ATP_ ion channels develop loci of hypercontracted sarcomeres during fatiguing exercise and have a faster rate of fatigue than animals with normal K_ATP_ ion channels [42]. However, their vulnerability to develop TrPs has never been examined.

### 3.2. Failure to Control Quantal ACh Release Leading to Endplate Noise

Endplate noise is a long-lasting phenomenon in TrPs, characterized by low amplitude, extremely high-frequency, and electrical activity in the resting state. It is indicative of an unusually high concentration of ACh at the motor endplate. ACh is released from the MNT in three different ways: (1) evoked quantal release, (2) spontaneous quantal release, and (3) spontaneous non-quantal release [43]. Evoked quantal release occurs following an efferent nerve impulse that results in a motor action potential, clearly not the case in resting muscle. Spontaneous non-quantal release (NQR) of ACh makes up 90–98% of the resting muscle total release of ACh and about half of that is from motor nerve endings. The NQR of ACh results in postsynaptic concentrations that are too low to evoke membrane responses unless AChE is inhibited [44], but can hyperpolarize the postsynaptic membrane [43]. It is independent of quantal release, and it can cause the depolarization of the postsynaptic membrane, as seen by occasional MEPPS [20]. It is an unlikely candidate for the cause of EPN because of its low concentration and because an MEPP is most commonly caused by the release of a single quantum of ACh [20]. Additionally, the depression of EPN activity by botulinum toxin strongly indicates that ACh is released from intracellular vesicles [21] rather than from leakage. Therefore, the most likely cause of EPN at the TrP is spontaneous quantal release (SQR) that occurs at rest. Spontaneous quantal release can cause subthreshold membrane depolarization that induces segmental sarcomere contraction both in skeletal and cardiac muscle [45]. Inhibition of AChE in the motor endplate zone, and expansion of AChRs beyond the motor endplate zone have an amplification effect of increasing ACh at the endplate, resulting in subthreshold membrane depolarization or a fully propagated motor action potential.

An animal model of TrPs created by blunt trauma to muscle followed by intensive exercise showed elevated ACh content and lowered AChE at the TrP spot, but the spontaneous electrical activity in their model consisted of positive sharp waves, fibrillation potentials, and fasciculation potentials [28], characteristic of denervation, rather than of TrPs.

Noncanonical signaling in which the nAChR itself is the signaling molecule [46] mediates changes in the quantal content of ACh. This modulatory mechanism of regulating changes in quantal content should increase the release of ACh molecules in response to blocking nAChRs. This has not been examined in TrP models.

### 3.3. Sympathetic Nervous System Contribution

The ⍺-adrenergic sympathetic nervous system (SNS) plays a major role in upregulating the effect of ACh at the NMJ in TrPs, as shown by the inhibiting effect of ⍺-adrenergic inhibitors, such as phentolamine on EPN. It achieves this either by increasing the release of ACh from the MNT or by otherwise increasing the concentration of ACh molecules at the motor endplate. The SNS innervates skeletal muscle and the NMJ both pre- and postsynaptically [47,48,49]. Elimination of the SNS influence on skeletal muscle by sympathectomy results in a decrease in MEPP frequency and amplitude by a presynaptic decrease in the quantal release of ACh, a reduction in the complexity of the NMJ, and postsynaptically, a reduction in the number of AChRs. Sympathomimetic drugs, such as epinephrine and norepinephrine reverse the effects of sympathectomy and improve NMJ transmission. Epinephrine and norepinephrine both decrease the spontaneous release of ACh quanta, an effect blocked by ⍺-adrenergic inhibitors in the case of epinephrine [49] and β-adrenergic inhibitors in the case of norepinephrine [50]. Alpha-2 agonists increase the MEPP frequency, an action blocked by the ⍺-adrenergic inhibitors phentolamine, prazosin, and yohimbe, indicating that the effect is presynaptic [51]. Synchronization of ACh release increases the likelihood of producing an action potential, an effect blocked by both alpha-2 and beta-2 agonists [45]. Epinephrine and other sympathomimetic agents modulate ACh quantal release via presynaptic P/Q calcium channels, and via N-type calcium channels postsynaptically to modulate intracellular [Ca^2+^]_c_. Β-adrenergic agonists act postsynaptically to enhance muscle contraction force. The SNS also acts indirectly postsynaptically via G-protein ⍺_i2_ and other genes to regulate intracellular [Ca^2+^]_c_ [47,48].

### 3.4. Adenosine Receptor Interaction with Muscarinic Receptors

Adenosine receptors A_1_ and A_2A_ at the MNT interact with M1 and M2 muscarinic receptors to regulate ACh through a feedback mechanism that responds to the frequency of action potential discharges at the NMJ and to levels of ATP and adenosine [51,52]. This mechanism serves as a detector of low concentrations of ATP associated with intense muscle activity. ATP concentration is higher at lower frequencies of membrane depolarization and lower at higher frequencies. The M1 faciliatory receptor is activated at lower concentrations of ACh with lower frequencies of firing, but higher concentrations of ACh associated with high frequency discharges activates inhibitory M2 receptors via A_2A_-adenosine receptors [53]. This is a protective mechanism that could limit the depletion of ATP that could impair intracellular Ca^2+^ reuptake at higher frequencies of motor action potentials.

### 3.5. Summary of Sympathetic Nervous System Effects

The SNS acts presynaptically to modulate the spontaneous ACh quantal release through the action of catecholamines. Quantal release is further modulated by the interaction of adenosine receptors A_1_ and A_2A_ with muscarinic receptors M1 and M2. Postsynaptic modulation of excitation–contraction coupling includes G-protein second messenger and protein kinase C pathways that regulate calcium influx into the muscle cytosol.

### 3.6. Brain-Derived Neurotrophic Factor

Brain-derived neurotrophic factor (BDNF) acts presynaptically to modulate ACh quantal release via protein kinase C pathways, regulated by a feedback control mechanism related to evoke (efferent nerve stimulation) muscle contraction [54]. The studies of BDNF in muscle pain have mostly been directed towards its role in nociception [55,56,57].

### 3.7. Muscle Fatigue

Muscle fatigue from intensive or repetitive muscle action may initiate a TrP that can persist after the initiating event. Muscle fatigue can involve multiple pathways involving muscle, but two systems of potentially particular relevance to TrPs and fatigue or weakness associated with TrPs will be discussed. The evoked quantal release of ACh leads to the process of excitation–contraction coupling, whereby the release of a neurotransmitter (ACh) is converted to an electrical signal through membrane depolarization and then to a mechanical act, muscle contraction. Membrane depolarization causes an electrical impulse to travel through the transverse t-tubule, resulting in the dephosphoralization of the dihydropyridine (DHPR) ion channel that then activates the RyR1 ion channel and releases Ca^2+^ from the sarcoplasmic reticulum (SR) into the muscle cytosol, where Ca^2+^ binds with troponin to initiate actin-myosin cross bridging and muscle contraction. Factors that affect the force and duration of muscle contraction and produce fatigue include those that lead to a motor unit action potential and those that modulate the level of intracellular Ca^2+^.

Muscle fatigue occurs after high intensity exercise. The time after one action potential is generated before a second action potential can be generated is prolonged in in vivo muscle contractions-to-fatigue studies under partially depolarized conditions in rat fast twitch skeletal muscle [58]. This phenomenon, called reprime time, is dependent on a membrane repolarization greater than −65 mV. Based on observations that muscle contractions can increase S-glutathionylation of Na^+^-K^+^-ATPase, disulfide was added between glutathione and oxidized cysteine residues, an action that can be mediated by reactive oxygen species. This inhibits Na^+^-K^+^-ATPase activity in the t-tubule system, essential for the maintenance of the Na^+^ and K^+^ electrochemical gradients during and after high intensity contractions. However, intense muscle activity also phosphorylates phospholemman (PLM) and dissociates it from Na^+^-K^+^-ATPase, which otherwise suppresses its activity. The effect of PLM dissociation from Na^+^-K^+^-ATPase is an increase in its activity. The two effects of intense muscle contraction balance each other, maintaining membrane gradient stability. However, when ATP levels are low, S-glutathionylation of Na^+^-K^+^-ATPase occurs, as in intense muscle activity, a persisting effect after fatiguing muscle stimulation. The increased inhibition of Na^+^-K^+^-ATPase could contribute to a decrease in t-tubule excitability, that along with lowered ATP levels, would cause a decrease in the force of muscle contraction. In fact, ATP levels can fall quite low after exercise [59]. An alternative explanation for a decreased Na^+^-K^+^-ATPase is that it is the result of lowered glycogen levels in muscle after muscle contraction, as ATP is generated via the glycolytic pathway [60]. A longer repriming period is in fact related to decreased Na^+^-K^+^-ATPase activity [61]. The fatigue rate is faster in K_ATP_-ion channel-deficient muscle fibers than in normal fibers, despite the fact that K_ATP_-ion channel opening increases with the rate of fatigue [30], so that it might be expected that the absence of K_ATP_ ion channels would not result in an altered rate of fatigue. However, the greater rate of fatigue may be due to fiber damage and contractile dysfunction in intensely exercised K_ATP_-deficient mice [62].

### 3.8. Ion Channelopathy

Muscle overload (muscle activity to or beyond fatigue) is a commonly accepted, though unproven, cause of TrPs, whether acute overload or chronic and repetitive. If this is the case, perhaps even elite athletes should have TrPs. In fact, there is literature related to the treatment of athletes with TrPs; (see for example [63,64,65,66]). However, there are no epidemiological studies of TrPs in athletes, whereas there are studies of myalgia following exercise or repetitive activity. One study of trigger point morphology incidentally showed taut bands in all studied keyboard workers whether or not they had myalgia [17].

Muscle overload or intense exercise can produce muscle injury or muscle cell apoptosis from an excess of [Ca^2+^]_c_. Mechanisms that protect muscle from such cellular injury and apoptosis include the K_ATP_ ion channel opening that limits Ca^2+^ ingress into the muscle cytosol when intense muscle activity lowers ATP levels. Another mechanism is the ryanodine receptor (RyR) ion channel for the egress of calcium from the SR into the cytosol and the ATP-dependent sarcoplasmic reticulum calcium ATPase (SERCA) mechanism for calcium reuptake from the muscle cytosol. Each of these molecular entities is known to have multiple mutations/variants that lead to either functional up- or downregulation, or to alter other channel functions as SERCA does for the Piezo protein mechanotransduction channels [67]. Molecular subunit alleles in ion channels not only up- or downregulate channel function, but can also alter sensitivity to inhibitory factors, as happens with K_ATP_ ion in the Cantú syndrome, a rare multisystem disorder of hypertrichosis, cardiomegaly, and skeletal and other anomalies [68].

#### 3.8.1. Ryanodine Receptor Channelopathy

The RyR regulates the influx of calcium from the intracellular SR stores into the myofibril cytosol. Ingress of calcium from the SR is accomplished largely through the RYR1 ion channel, the predominant isoform in skeletal muscle. The amount of calcium that enters and leaves the cytosol is critical, for too little ionized calcium reduces the force of muscle contraction, and too much calcium can lead to muscle cell injury or cell death. Removal of calcium from the cytosol is accomplished by the SERCA mechanism. The RyR is the largest of the known ion channels. It is a P-type gated channel that has six transmembrane regions. It opens in response to membrane depolarization, part of the excitation–contraction coupling process, and is also a calcium induced calcium channel. Membrane depolarization activates L-type voltage-gated Ca^2+^ DHPR channels, that otherwise physically block RyR1 channels to prevent Ca^2+^ transit through the receptor pore. Ca^2+^ in the cytosol binds to troponin, removes tropomyosin from myosin-head binding sites, allowing cross bridging with actin. Repeated binding, release, and binding allows for a progressive ‘walking’ of actin along the myosin molecule, shortening the sarcomere. RyR1 activity is modulated by many factors, including phosphorylation, oxidation, nitrosylation, and mutation. Several muscle diseases are associated with RyR1 mutations, including central core and multicore diseases, but the most relevant one for TrPs is malignant hyperthermia via a gain of function mutation.

#### 3.8.2. Malignant Hyperthermia

Malignant hyperthermia (MH) is a genetic disorder of a potentially lethal hypermetabolic crisis associated with a rapid, uncontrolled ingress of Ca^2+^ into muscle cells [69]. The RyR1 gene is the key mutant allele, most often inherited as an autosomal dominant. There are over 400 mutants of the RyR1 gene, of which more than 40 are associated with MH. The genes CACNAIS and STAC3 are also associated with MH [69]. The clinical syndrome includes muscle rigidity and rhabdomyolysis. The MH mutations result in a ‘leaky’ RyR1 ion channel that results in a rapid and uncontrolled transit of Ca^2+^ from the SR into the cytosol, causing persistent actin-myosin cross bridging and sarcomere shortening. Ca^2+^ must be removed from the cytosol to reverse actin-myosin cross bridging and to allow for the muscle to relax. As the ATP essential for Ca^2+^ reuptake is depleted, Ca^2+^ reuptake fails and an excess of cytosolic Ca^2+^ accumulates in the cytosol, resulting in persistent actin-myosin cross bridging, tremors, contractions, and rigidity.

Phenotypic penetrance is variable and incomplete in MH. The clinical syndrome is triggered by exposure to certain substances, such as succinylcholine and halothane. RyR1 ion channel function is modulated by other substances or processes, including Ca^2+^, Mg^2^, caffeine, ATP, calmodulin, phosphorylation, and oxidation [70,71]. Arrythmias, seizures, and myopathy can occur. However, when not exposed to a trigger, the individual with the RyR1 mutation can function perfectly well and be healthy. Gene penetrance is incomplete and multiple different alleles can exist, making clinical presentation variable. Thus, the potential exists that an RyR1 allele might create a vulnerability to develop TrPs when the carrier is exposed to intense exercise or to another trigger.

#### 3.8.3. RyR Mutation or Post-Translational Modification Causing Exercise Intolerance

That RyR1 functional variants can be relevant to TrP myofascial pain syndromes is suggested by the report of 14 individuals with exertional rhabdomyolysis or myalgia. Thirty-nine individuals who were evaluated had one of nine mutations or variants. Most had no or only subtle weakness. Some had affected family members as well but without rhabdomyolysis or exertional myalgia. Some were able to engage in intense exercise [72]. Remodeling of the RYR1 complex that is the major Ca^2+^ release channel in skeletal muscle by progressive PKA-hyperphosphorylation, S-nitrosylation, and by depletion of the RyR1 stabilizing subunit calstablin results in leaky channels that cause decreased exercise tolerance [73]. RyR1 is post-translationally modified in heart failure patients leading to pathological Ca^2+^ release as a potential mechanism for skeletal muscle weakness and impaired exercise tolerance [74]. Thus, it is possible that either by mutation or by post-translational modification of the RyR1 ion channel that is either genetic or secondary to intense exercise or by some other means, an individual may have significantly diminished exercise tolerance and be more vulnerable to a post-translational modification of the ion channel that would be either muscle specific or fiber-type specific.

#### 3.8.4. ATP Dependent Channel Mutations and Muscle Function: The K_ATP_ Ion Channel (Figure 1)

The K_ATP_ ion channel opens when intense exercise lowers the concentration of ATP that is required for the maintenance of the Na^+^K^+^ pump. The Na^+^K^+^ pump is necessary for sarcolemma and t-tubular action potentials, the generation of actin-myosin cross bridging, and the SR-ATPase that powers the Ca^2+^ reuptake into the sarcolemma [75]. Intense muscle exercise consumes ATP, resulting in very low ATP levels [42,75]. The ATP-depleting effect of intense muscle exercise differs among muscle phenotypes and slow and fast twitch muscle fibers. ATP levels during normal activity rarely fall below 60–70% of pre-exercise levels. Muscle protective mechanisms, as described previously, prevent muscle activity from continuing at such a high intensity as to cause ATP levels to fall dangerously low. One such mechanism is the ATP-potassium channel, particularly the K_ATP_ ion channel that when open limits the influx of Ca^2+^ into the cell. The K_ATP_ ion channel, located in the t-tubule, serves as an energy and pH sensor. The ion channel is made of four subunits: the Kir subunits that are inwardly rectifying K^+^ channel-forming subunits and two sulfonylurea receptor (SUR) regulatory subunits. The subunits are gene-encoded for Kir6.1 and Kir6.2, such as the KCNJ8 gene that encodes Kir6.2, and by ATP binding cassette genes ABCC8 (SUR1) and ABCC9 (SUR2) [74]. In normal circumstances, including continuous low-level exercise, the myoprotective K_ATP_ ion channel is closed. When ATP levels fall significantly, the K_ATP_ ion channel opens. It also opens when the pH falls and when there is metabolic stress, such as ischemia and hypoxia. The result of opening the channel is the conservation of ATP. Outward K^+^ currents oppose inward Na^+^ currents that depolarize the muscle membrane when the ion channel is open, maintaining resting membrane potentials, resulting in a decreased action potential amplitude, reduced release of Ca^2+^ from the SR into the muscle cytosol, decreased Ca^2+^-ATPase activity, and therefore less ATP consumption [30,75].

In K_ATP_ knockout mice (K_ATP_ −/−) or in muscle treated with a K_ATP_ inhibitor, such as glibenclamide, the myoprotective action of K_ATP_ is absent. Under usual circumstances, such knockout mice behave normally and show normal muscle function. However, when muscles lacking K_ATP_ ion channel activity are intensively exercised to fatigue, as when muscle is subjected to high-frequency tetanic stimulation, ATP levels fall and [Ca^2+^]_c_ rise. The muscle membrane becomes hyperdepolarized and muscle fibers may become super contracted [30,42,75,76]. Peak tetanic force is reduced by almost a third in K_ATP_ knockout (Kir6.2 −/−) mice compared to wild type mice. The Kir6−/− animals recover only about 2/3 of their peak tetanic force after 15 min. Thus, the Kir6−/− knockout mice are less able to tolerate intense exercise, may develop an energy crisis, and are more vulnerable to contractile dysfunction [40,75,77] (Figure 1). Verapamil significantly reduced the high levels of [Ca^2+^]_c_ in these circumstances [42,77].
Figure 1The K_ATP_ ion channel effect on muscle function under normal circumstances and when deficient or absent.
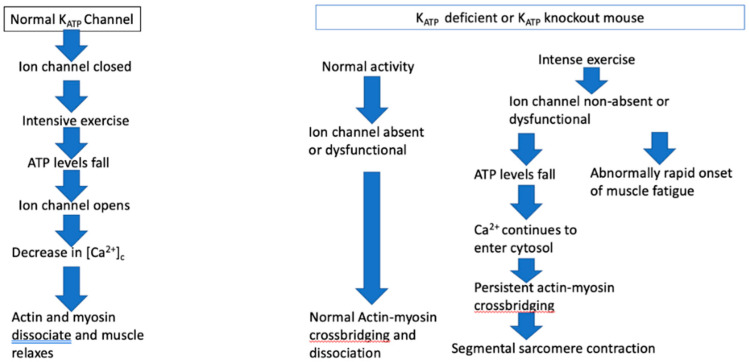


K_ATP_ ion channel activity is phenotype dependent and muscle specific; its activity is greater in fast twitch muscle than in slow twitch oxidative fibers that have a greater capacity to generate ATP, and that are least affected by the absence of K_ATP_ ion channels. Fast twitch glycolytic fibers are most dependent on the myoprotective effect of the K_ATP_ ion channel [74]. Rat soleus muscle is a slow twitch muscle, whereas the extensor digitorum longus (EDL) is a mixed slow and fast twitch muscle, and the rat flexor digitorum brevis is predominantly fast twitch [76]. The two fiber types are compartmentalized in the rat EDL, the medial compartment comprises primarily of slow twitch oxidative fibers and the lateral compartment of predominantly fast twitch glycolytic fiber. The medial compartment has greater fatigue resistance than the lateral compartment [77]. Thus, it is critical to understand the nature of the muscle and its fiber types when conducting studies of K_ATP_ function and of muscle fatigue.

Muscle lacking functioning K_ATP_ activity resemble TrP containing muscle in several ways. Muscle is unaffected when activity is moderate and not excessive or intensive. This is consistent with the normal human skeletal muscle function until it is overloaded or worked beyond its capacity. Not all muscles or muscle fibers are affected. In TrP affected muscle, the involvement of muscle fibers is heterogeneous, not homogeneous. Segmental sarcomere contraction may be seen in TrP regions, such as the super contraction of muscle fibers in K_ATP_ knockout mice that are subjected to muscle overload. Muscle force is eventually reduced, and the onset of fatigue is faster in knockout mice. In humans, muscles with TrPs are weaker than normal, as shown by a rapid return of normal strength after the TrP is inactivated, though the mechanism may be entirely different from that in K_ATP_ knockout mice.

## 4. Conclusions

The new hypothesis proposes that in at least a subset of situations in which TrPs form the initiating event is either an acute muscle overload or repetitive muscle action to fatigue in which muscle activity performed beyond the sustainable capacity of the muscle results in either an excess of ACh molecules at the motor endplate or a dangerously high concentration of Ca^2+^ in the muscle cytosol or both. In either case, there is a potential danger of muscle injury or damage. The new hypothesis postulates that there are feedback mechanisms both at the presynaptic and the postsynaptic levels to prevent either a consequent dangerous drop in levels of ATP or a dangerous rise in intracellular calcium levels, and that one or more of the myoprotective feedback mechanisms fail.

The new hypothesis suggests areas of productive research and potential management regimens that could be explored. For example, genetic studies of individuals with persistent or recurrent myofascial trigger point pain syndromes could identify ion channelopathies, that could lead to simple treatment measures, such as the avoidance of caffeine, or employ alpha or beta blockers, CGRP inhibitors, or modified exercise regimens. Furthermore, studies of fatiguability in TrP-containing muscles could explain some clinical features of TrPs.

## Data Availability

Not applicable.

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
