# Peer review of "A New Unified Theory of Trigger Point Formation: Failure of Pre- and Post-Synaptic Feedback Control Mechanisms"

_ijms, 2023, doi:10.3390/ijms24098142_

Round 1

Reviewer 1 Report (New Reviewer)

Thank you for submitting this interesting article.  The paper presented a new theory for the origin of the myofascial trigger point to replace the integrated hypothesis. The manuscript provides a review of content for the subject area.

The topic is not original because there are some previous investigations about histology which provide a theory: the integrated hypothesis. But it is relevant because it is necessary to have a good foundation to describe de TrP. There is a lot of controversy about TrP, so a new approach which includes the brain is important.

They do not provide the search formula which is done to redact the paper. It would be good to have more expert’s opinion like: Orlando Mayoral or Manel Santafé.

Based on the evidence, the paper provides a final conclusion which summarizes the paper.

The references are appropriate, there is not so much investigation about this. They consider it, and particularly, the most important research groups.

Additional comments:  There are some unnecessary underlined in the figure 1 (dissociate, crossbridging).

Reviewer 2 Report (Previous Reviewer 3)

All suggestions have been carried out. I think the article has improved a lot.

This manuscript is a resubmission of an earlier submission. The following is a list of the peer review reports and author responses from that submission.

Round 1

Author Response

Could reviewer 2 please give me some help in revising my paper by giving me some specifics as to what the reviewer thought was lacking in the paper, since the reviewer said that most aspects of the paper must be improved. However, the reviewer gave no instances or examples of shortcomings of the paper. To reviewer 1, I have added some clinical relevance at the end of the paper, in terms of adding some treatment implications. 

To reviewer 2 I extensively revised the manuscript. 

I appreciate the reviewer noting errors in spelling and punctuation and I have cleaned up the text to clarify and make it easier to understand. 

Re: I have added to the introduction to make it more relavant and pertinent and to make it clear why I think that this paper is necessary. 

Re: appropropriate research design and methods descrived: this is a concept paper, not a laboratory or clinical research paper. I have tried to give the background on which my thinking is based, and then to detail the way each of the specified mechanisms can work to produce trigger points. I have tried to clarify that in the revision. I hope that the reasoning is clear. 

Re: Results and conclusions. I have also revised those sections so that they reflect the physiologic basis on which I built my hypothesis. I hope that these sections are clear. 

Re: Methods and results: because this is a concept paper, the methodology is really a synthesis of the literature in order to develop a new concept. The results are really extracting the relevant findings from the literature in order to come up with a new concept. I think that is clear, in that I have detailed what is known about trigger points that is relevant, then what is known about the sympathetic nervous system and about certain channelopathies that are relevant to the new hypothesis, and then put them together to come up with a concept that one or more different feedback control mechanisms would have to fail in order to produce a chronic trigger point. 

I thank the reviewer for the opportunity to consider the problems the reviewer raised and to try to address them. 

I have uploaded a clean copy of the revisions. Below, I am adding some of the sections with more important changes that I made.  I have italicized the changes.  There are other smaller changes in the manuscript that I did not copy here. 

Citation: Gerwin, R.D. A New Unified Theory of Trigger Point Formation: Failure of pre-and post synaptic feedback control mechanisms.  2023, 22, x. https://doi.org/10.3390/xxxxx

Academic Editor(s):

Received: date

Accepted: date

Published: date

Publisher’s Note: MDPI stays neutral with regard to jurisdictional claims in published maps and institutional affiliations.

Copyright: © 2023 by the authors. Submitted for possible open access publication under the terms and conditions of the Creative Commons Attribution (CC BY) license (https://creativecommons.org/licenses/by/4.0/).

Johns Hopkins University, School of Medicine, Baltimore, MD, USA

  1. Introduction

The myofascial trigger point (TrP) is considered to be the underlying cause of the myofascial pain syndrome (MPS). Described by Drs. Janet Travell and Seymore Rinzler in 1946 [1], it has been increasingly studied over the subsequent years, but the mechanism(s) that initiates the TrP has never been adequately described. I propose a New Hypothesis about for the initiation of the TrP. In doing so, I acknowledge that the TrP, like many other dysfunctional conditions, may have more than one cause, all leading to a common, final, entity, the TrP. Thus, the proposals presented herein may be relevant to a majority of TrPs, or only to a subset of them.

Most studies of the TrP treat it as an established entity; i.e., address the TrP as it already exists, or address its consequences, such as referred pain or central sensitization.  Rarely do they address the mechanisms that lead to TrP formation. Travell and Simons discussed perpetuating factors that are relevant to a person’s vulnerability toward MPS, but that does not address specifically the actual mechanism of TrP formation [2]. Simons’ original Integrated Trigger Point Hypothesis (the Integrated Hypothesis) attempts to do that. It starts with an energy crisis component based on several fundamental observations:

line 58 ff: based on a synthesis of observations from many pubished studies from different authors. This New Hypothesis is offered as a means of stimulating research into the physiology of TrP formation, but it also has implications for the mechanism of fatigue and weakness in muscles with trigger points, and also for potential treatment paradigms for those individuals with chronic MPS.

Myofascial TrPs have been were described for about 75 years as ago by Travell and Rinzler described when they reported non-cardiac angina-like chest pains in 1946 [1]. Many of the subsequent publications were have been descriptive, presenting clinical findings

line 96 ff: mechanisms will be described in detail, leading to a hypothetical basis of TrP formation that may lead to additional therapeutic measures and help direct future research as outlined at the end of this presentation, and that ultimately may add to the treatment possibilities for MPS..  

line 142ff: 

were on established TrPs and may not speak directly to the situation during the development of the TrP [13]. A  The CGRP effect at the NMJ is delayed not immediate, but it may be a very early downstream result of an earlier initiating event, and may contribute to the development of the TrP by increasing the number of ACh molecules at the endplate zone.

Neurotransmitters and cytokines present at high concentration in the extracellular TrP milieu, such as CGRP and substance P, may also produce neurogenic edema. The h

line 298 ff: 

operative at the TrP that has foci of very high frequency electromyographic discharges. Its relationship to the SNS is, at least in part, to moderate the effect of a stress-related SNS response that results in an increase in catecholamine release and a catecholamine-induced increase in ACh release from the NMJ.

3.4. Summary of sympathetic nervouse system effects

In summary, the SNS action at the TrP includes an ⍺-adrenergic-related presynaptic upregulation of ACh release from the MNT. Candidates for the specific site of action include the adenosine receptors  A1 and A2A,. Failure of a feedback control mechanism in this case would be in the interplay between the presynaptic adenosine and muscarinic receptors.  Post-synaptic modulation is primarily related to mediated by the β-adrenergic regulation 

line 516ff:

Trigger points produce a non-neurogenic muscle weakness that is rapidly reversed with inactivation of the TrP, but it would interesting to see if some of the weakness could be related to ion channelopathy-related muscle fatigue.

line 555ff:

Implications for Clinical Practice:

  1. If a KATP ion channelopathy is found, we know that muscle function is not stressed during normal activity, but cannot tolerate intensive exercise. The patient could be instructed to moderate activity so as to avoid untolerated stress. In RyR1 channelopathies, avoidance of something as simple as caffeine may reduce TrP formation.
  2. If a sympathetic dysfunction is suspected, an alpha- or beta- adrenergic antagonist, or a CGRP antagonist could be tried to ameliorate the development of trigger points.
  3. Localized tetantic-induced fatigue studies could be used in particularly difficult cases to identify an ion channelopathy and to prescribe appropriate activity limits to minimize further TrP formation.
  4. In a number of these possibilities, low level exercise may be tolerated whereas high-intensity exercise may lead to muscle stress and TrP formation.

line 595 ff:

  1. The effect of BDNF on non-evoked muscle contraction should be studied to see if it is active in resting muscle, thereby possibly implicated in TrP formation.

Reviewer 2 Report

I have read with great interest the current paper . The genesis of myofascial trigger points is likely multifactorial. the author I propose a New Hypthesis  about the intitation of the TrP. In doing so, he acknowledges that the TrP,  may have more than one cause, all leading to a common, final,  entity. The author presented  a sound scientific theory based on imaging and other factors that gives merit to researchers of basic science and less merit to clinicians . As a comment I would suggest writing the potential  applicability

of this theory to clinicians . the interest to the reader depends who is the reader and in what field(clinical or basic science )  he/she involves . 

Typo  error in line 484 ot- should be of at the end of the line . 

Author Response

To reviewer 2 I extensively revised the manuscript. 

I appreciate the reviewer noting errors in spelling and punctuation and I have cleaned up the text to clarify and make it easier to understand. 

Re: I have added to the introduction to make it more relavant and pertinent and to make it clear why I think that this paper is necessary. 

Re: appropropriate research design and methods descrived: this is a concept paper, not a laboratory or clinical research paper. I have tried to give the background on which my thinking is based, and then to detail the way each of the specified mechanisms can work to produce trigger points. I have tried to clarify that in the revision. I hope that the reasoning is clear. 

Re: Results and conclusions. I have also revised those sections so that they reflect the physiologic basis on which I built my hypothesis. I hope that these sections are clear. 

Re: Methods and results: because this is a concept paper, the methodology is really a synthesis of the literature in order to develop a new concept. The results are really extracting the relevant findings from the literature in order to come up with a new concept. I think that is clear, in that I have detailed what is known about trigger points that is relevant, then what is known about the sympathetic nervous system and about certain channelopathies that are relevant to the new hypothesis, and then put them together to come up with a concept that one or more different feedback control mechanisms would have to fail in order to produce a chronic trigger point. 

I thank the reviewer for the opportunity to consider the problems the reviewer raised and to try to address them. 

I have uploaded a clean copy of the revisions. Below, I am adding some of the sections with more important changes that I made.  I have italicized the changes.  There are other smaller changes in the manuscript that I did not copy here. 

Citation: Gerwin, R.D. A New Unified Theory of Trigger Point Formation: Failure of pre-and post synaptic feedback control mechanisms.  2023, 22, x. https://doi.org/10.3390/xxxxx

Academic Editor(s):

Received: date

Accepted: date

Published: date

Publisher’s Note: MDPI stays neutral with regard to jurisdictional claims in published maps and institutional affiliations.

Copyright: © 2023 by the authors. Submitted for possible open access publication under the terms and conditions of the Creative Commons Attribution (CC BY) license (https://creativecommons.org/licenses/by/4.0/).

Johns Hopkins University, School of Medicine, Baltimore, MD, USA

  1. Introduction

The myofascial trigger point (TrP) is considered to be the underlying cause of the myofascial pain syndrome (MPS). Described by Drs. Janet Travell and Seymore Rinzler in 1946 [1], it has been increasingly studied over the subsequent years, but the mechanism(s) that initiates the TrP has never been adequately described. I propose a New Hypothesis about for the initiation of the TrP. In doing so, I acknowledge that the TrP, like many other dysfunctional conditions, may have more than one cause, all leading to a common, final, entity, the TrP. Thus, the proposals presented herein may be relevant to a majority of TrPs, or only to a subset of them.

Most studies of the TrP treat it as an established entity; i.e., address the TrP as it already exists, or address its consequences, such as referred pain or central sensitization.  Rarely do they address the mechanisms that lead to TrP formation. Travell and Simons discussed perpetuating factors that are relevant to a person’s vulnerability toward MPS, but that does not address specifically the actual mechanism of TrP formation [2]. Simons’ original Integrated Trigger Point Hypothesis (the Integrated Hypothesis) attempts to do that. It starts with an energy crisis component based on several fundamental observations:

line 58 ff: based on a synthesis of observations from many pubished studies from different authors. This New Hypothesis is offered as a means of stimulating research into the physiology of TrP formation, but it also has implications for the mechanism of fatigue and weakness in muscles with trigger points, and also for potential treatment paradigms for those individuals with chronic MPS.

Myofascial TrPs have been were described for about 75 years as ago by Travell and Rinzler described when they reported non-cardiac angina-like chest pains in 1946 [1]. Many of the subsequent publications were have been descriptive, presenting clinical findings

line 96 ff: mechanisms will be described in detail, leading to a hypothetical basis of TrP formation that may lead to additional therapeutic measures and help direct future research as outlined at the end of this presentation, and that ultimately may add to the treatment possibilities for MPS..  

line 142ff: 

were on established TrPs and may not speak directly to the situation during the development of the TrP [13]. A  The CGRP effect at the NMJ is delayed not immediate, but it may be a very early downstream result of an earlier initiating event, and may contribute to the development of the TrP by increasing the number of ACh molecules at the endplate zone.

Neurotransmitters and cytokines present at high concentration in the extracellular TrP milieu, such as CGRP and substance P, may also produce neurogenic edema. The h

line 298 ff: 

operative at the TrP that has foci of very high frequency electromyographic discharges. Its relationship to the SNS is, at least in part, to moderate the effect of a stress-related SNS response that results in an increase in catecholamine release and a catecholamine-induced increase in ACh release from the NMJ.

3.4. Summary of sympathetic nervouse system effects

In summary, the SNS action at the TrP includes an ⍺-adrenergic-related presynaptic upregulation of ACh release from the MNT. Candidates for the specific site of action include the adenosine receptors  A1 and A2A,. Failure of a feedback control mechanism in this case would be in the interplay between the presynaptic adenosine and muscarinic receptors.  Post-synaptic modulation is primarily related to mediated by the β-adrenergic regulation 

line 516ff:

Trigger points produce a non-neurogenic muscle weakness that is rapidly reversed with inactivation of the TrP, but it would interesting to see if some of the weakness could be related to ion channelopathy-related muscle fatigue.

line 555ff:

Implications for Clinical Practice:

  1. If a KATP ion channelopathy is found, we know that muscle function is not stressed during normal activity, but cannot tolerate intensive exercise. The patient could be instructed to moderate activity so as to avoid untolerated stress. In RyR1 channelopathies, avoidance of something as simple as caffeine may reduce TrP formation.
  2. If a sympathetic dysfunction is suspected, an alpha- or beta- adrenergic antagonist, or a CGRP antagonist could be tried to ameliorate the development of trigger points.
  3. Localized tetantic-induced fatigue studies could be used in particularly difficult cases to identify an ion channelopathy and to prescribe appropriate activity limits to minimize further TrP formation.
  4. In a number of these possibilities, low level exercise may be tolerated whereas high-intensity exercise may lead to muscle stress and TrP formation.

line 595 ff:

  1. The effect of BDNF on non-evoked muscle contraction should be studied to see if it is active in resting muscle, thereby possibly implicated in TrP formation.

Reviewer 3 Report

This work considers muscle fatigue as the central axis of the pathophysiology of myofascial trigger points. The participation of new scientific findings in the neuromuscular field in the construction of this pathophysiological vision is suggested. The manuscript is well documented. However I have found some inaccuracies that will improve the rigor of the article and its understanding:

1.-Introduction is too long.

2.-Unfortunately there are still authors who oppose myofascial trigger points. In the general introduction it is necessary to comment on them even briefly.

3.-This is not exact: These feedback control mechanisms control the spontaneous, non-evoked, release of ACh from the motor nerve terminal (MNT) and therefore the frequency of motor action potentials, including miniature endplate potentials (MEPPS), and control the flow of Ca2+ through ion channels into the cytosol that results in the binding of actin to myosin leading to sarcomere contraction.

Better like this: These feedback control mechanisms control the spontaneous, non-evoked, release of ACh from the motor nerve terminal (MNT) and therefore the frequency of miniature endplate potentials (MEPPS), and at the postsynaptic level control the flow of Ca2+ through ion channels into the cytosol that results in the binding of actin to myosin leading to sarcomere contraction.

4.- Is the local spasm response not mentioned anywhere? There is abundant literature on this and the nervous system is involved. Doesn't the author think it should be debated?

2.1. Electrophysiology:

Lines 102-103: It is not quite correct. Normal resting muscle is electrically silent; the resting muscle electromyogram (EMG) should show little to no activity. EndPlate noise is recorded in the NMJ area of normal resting muscles. Plaque noise is how spontaneous neurotransmission is recorded by EMG. Spontaneous neurotransmission always exists.

See:

- Kimura J. Electrodiagnosis in Diseases of nerve and Muscle: Principles and Practices. 2nd ed. Philadelphia: FA Davis.1989.

-Liley AW. An investigation of spontaneous activity at the neuromuscular junctíon of the rat. J Physiol 1956a;132: 650–66.

-Van Putten MJ, Padberg M, Tavy DL. In vivo analysis of end-plate noise of human extensor digitorum brevis muscle after intramuscularly injected botulinum toxin type A. Muscle Nerve 2002;26:784–90. DOI: 10.1002/mus.10274

- Macgregor J, Graf von Schweinitz D. Needle electromyographic activity of myofascial trigger points and control sites in equine cleidobrachialis muscle--an observational study. Acupunct Med 2006;24:61–70.

In healthy muscles, end plate noise is low frequency, low amplitude, and difficult to find. On the contrary, in the MrP the end plate noise is high frequency, greater amplitude and easy to find. That is the reason for the mistake of many clinicians.

Lines 109-110:  ….. representing a failure of the feedback mechanisms that regulate the release of ACh from the motor nerve terminal. However, it may be other factors such as an increase in sympathetic activity (see Kah et al 2016 of reference 43)

2.3. Biochemical pathophysiology:

The hypoechoic appearance of the TB on high-definition ultrasound is consistent with neurogenic edema, but this has yet to be proven experimentally [21,22]. Can I suggest the existence of glycosaminoglycans? That has been shown by Margalef et al 2019.

Margalef R, Sisquella M, Bosque M, Romeu C, Mayoral O, Monterde S, Priego M, Guerra-Perez R, Ortiz N, Tomàs J, Santafe MM. Experimental myofascial trigger point creation in rodents. J Appl Physiol (1985). 2019 Jan 1;126(1):160-169. doi: 10.1152/japplphysiol.00248.2018.

2.4. Histopathology:

I propose "Histopathology: Current evidence"

2.5. Ultrasound imaging of Trigger Points

Elastography is not discussed??

3. Discussion:

This section is not a discussion. I propose as a title "Analysis of elements related to The New Trigger Point Hypothesis", or another similar title.

It seems that only muscular fatigue is proposed as the beginning of the pathophysiological proposal. It must be considered that:

1.- An increase in physical activity necessarily requires an evoked neurotransmission (it is the only way to have voluntary muscle activity). An intensive evoked neurotransmission implies an alteration of the MEPPs in two possible ways: if it is exhausting, we will have depletion of the synaptic vesicles and therefore a decrease in the frequency of the MEPPs. If it is intensive but not strenuous, we will have an increase in the frequency of MEPPs due to an increase in axoplasmic calcium.

2.-Intensive physical activity overflowing with protective mechanisms suggests that elite athletes or exhaustive physical activity in amateur athletes should always cause myofascial trigger points. It also seems that it would be the only way for these to appear.

Taking this into account, perhaps this section should be explained using other words.

Line 190: 2) the postsynaptic ionotropic potassium and sodium channels that control influx of Ca2+ into the muscle citosol

It's not exact. With spontaneous neurotransmission, postsynaptic nAChRs allow Na entry at subthreshold levels for the firing of the muscle action potential. However, enough sodium enters to activate the ryanodine receptors of the sarcotubular system, thus leaving large amounts of calcium.

3.2. Failure to control ACh quantal release from the motor nerve terminal

Line 219. Remember that the electrical silence of neuromuscular synapses does not exist when recording with needle EMG.

Line 222: Not exact: Non-quantum or subquantum release is different from MEPPs. Each MEPPs releases a synaptic vesicle, that is, a quanta. Delete “.., non-quantal,..”

3.3. Sympathetic Nervous Systerm Contribution

Table 2 and 3. Although they highlight the importance of sympathetic innervation, they are redundant with respect to the text.

Line 241-244. It would be more accurate to note that the consequences of sympathectomy are obtained after several weeks.

3.4. Summary of sympathetic nervouse system effects

LIne 286: “G protein-related secondary messenger protein kinase C pathways that

govern actin-myosin coupling and sarcomere contraction” the concept of PKC has not yet been introduced.

3.7. Ion Channelopathy

Line 364: perhaps it should be explained what the Cantu Syndrome is.

3.7.2. Malignant hyperthermia

It is a very innovative and daring proposal.

4. Conclusion and Implications

4.1. Implications for Clinical Practice:

Stress or muscle tension from incorrect postures: which of the 4 proposals would you enter?

Some words could be written differently:

Line 156: bupivicaine is better as bupivacaine

Ryanadine (lines 186 and 371) is better known as Ryanodine

Author Response

Response to Reviewer 3:

I thank the reviewer for the careful reading of the manuscript that is evidenced by the reviewer’s comments, and for the reviewer’s recommendations.

  1. The introduction has been reduced in length by removing redundancies and irrelevant comments.
  2. Unfortunately there are still authors who oppose myofascial trigger points. In the general introduction it is necessary to comment on them even briefly.

Reply: I have noted the controversey,  and included a reference to the most recent challenge. “The myofascial trigger point (TrP), considered as the underlying cause of the myofascial pain syndrome, was described by Travell and Rinzler in 1946 [1].  It has been increasingly studied years since, despite a controversy over its existence [2][Quintner et al 2015].

3. This is not exact:

Reply: I have rewritten the sentence taking the reviewer’s suggestion and I thank the reviewer for pointing out an error in my characterization of ACh release.

The postulated dysfunctional feedback mechanisms include those that regulate the release of spontaneous non-evoked quantal ACh at the NMJ that induce miniature endplate potentials and those that modulate the release of ionized calcium into the muscle cell that results in the binding of actin to myosin leading to sarcomere contraction, and the reuptake of ionized calcium from the muscle cytosol that reverses the process and leads to muscle relaxation.”

4. Local spasm is not mentioned

Reply: Local sarcomere contraction is not really the same as spasm/cramp. Muscle spasm/cramp is defined as an involuntary, abnormal muscular contraction, but is associated with repetitive firing of motor units potentials at high rates (up to 150/sec. (see Miller TM and Layzer R. Muscle Nerve 2005;32(4):431 and Swash M et al. Eur J Neurology 2019;26(2):214). I have tried to make it clear that the EMG at the trigger point is endplate noise and not motor action potentials, and referred to studies that showed that segmental sarcomere contraction can be the result of subthreshold membrane depolarization. Therefore, I did not open the question about muscle cramps/spasm not being trigger points. I think that discussion is a bit removed from the point that I am trying to make regarding feedback control systems regulating spontaneous ACh quantal release. If the reviewer is satisfied with that, I would rather not open that discussion and lengthen the paper more than it is.

2.1 Electrophysiology

Reply: I thank the reviewer for pointing out the error. I have rewritten the section to make it clear that resting muscle activity is relatively quiet, but that there is activity (MEPPs). I have also corrected the paper and stated that it is spontaneous rather than evoked quantal release of ACh, not non-quantal release, that gives rise to MEPPs and to endplate noise.

Normal resting muscle is relatively electrically quiet. Miniature endplate potentials (MEPPs) occurring at a frequency of 1-6 per second and endplate spikes as seen on electromyographic studies, occurs in resting muscle [19,20] [Liley, 1956, Vyskocil ref 39]. In contrast to resting muscle, the electromyogram (EMG) of the TrP shows persistent, low amplitude (5-50 ?V), high frequency activity that looks like high frequency MEPPs punctuated by intermittent, higher amplitude (100-600 ?V), initially negative, biphasic, endplate spikes [13]. Resting TrP EMG activity, termed endplate noise (EPN) may be as much as 2-3 orders of magnitude faster than normal MEPP frequency. “

and

“Endplate noise is a long-lasting, persistent phenomenon of TrPs, characterized by low amplitude, extremely high-frequency, electrical activity in the resting state in contrast to the usual rather quiet resting state electrical activity.  The high-frequency TrP activity indicates that there is an unusually high concentration of ACh at the motor endplate. ACh is released from the MNT in three different ways: 1) evoked quantal release, 2) spontaneous quantal release, and 3) spontaneous non-quantal release [43][Tyapkina et al 2013].  Evoked quantal release occurs following an efferent nerve impulse that results in a motor action potential, clearly not the case in muscle at rest. Spontaneous non-quantal release (NQR) of ACh makes up 90-98% of the resting muscle total release of ACh and about half of that is from motor nerve endings. The NQR of ACh results in postsynaptic concentrations that are too low to evoke responses unless AChE is inhibited [44] [Nassenstein et al 2015], can hyperpolarize the postsynaptic membrane [43][Tuapkina et al 2012], and is independent of quantal release [19][19](Vyskocil ref 39]. It can cause depolarization of the postsynaptic membrane as seen by occasional MEPPS [19][Vyskocil ref 39].  It is an unlikely candidate for the cause of EPN because of its low concentration and because a MEPP is most commonly caused by the release of a single quantum of ACh [19][Vyskocil ref 39]. Additionally, the depression of EPN activity by botulinum toxin strongly indicates that ACh is released from intracellular vesicles [21][Kuan et al, 2002].  The most likely cause of EPN at the TrP therefore is spontaneous quantal release (SQR) that also occurs at rest. Spontaneous quantal release can cause subthreshold membrane depolarization that induces segmental sarcomere contraction both in skeletal and cardiac muscle [45].  Inhibition of AChE in the motor endplate zone, and expansion of AChRs beyond the motor endplate zone can have the same effect of increasing the effect of ACh at the NMJ resulting in subthreshold membrane depolarization or to a fully propagated motor action potential. “

2.3 Biochemical pathophysiology

Reply: I cited the findings in the Margoleff study that found GAG at the trigger point (they did not mention that GAG was found along the taut band outside of the trigger point) and mentioned it as an alternative possibility.

“The hypoechoic appearance of the TB on high-definition ultrasound is consistent with neurogenic edema, although there are other explanations for the nodular swelling at the TrP like the recent finding of glycosaminoglycans surrounding contraction knots in an experimental TrP paradigm [24,25]. “

3. Discussion

Reply: I agree that the section is best renamed and I took the reviewer’s suggestion in renaming it.

Fatigue: I mention fatigue only because it is so commonly cited as a cause. However, muscle overuse, which really means muscle activity to the point of fatigue or beyond, occurs in eccentric lengthening, in psychological stress when muscle tightens, in occupational and postural stresses, and even in metabolic and nutritional disorders like hypothyroidism (a hypometabolic state) and iron deficiency that creates a hypometabolic situation. There, in many if not most of the predisposing situations, fatigue is a final common pathway to the development of TrPs.

Fatigue is the primary factor that leads to the development of the trigger point. Based on your comment, which I agree can be misinterpreted, I have clarified the point that spontaneous quantal release of ACh is responsible for endplate noise and that evoked quantal release occurs in muscle activity that leads to fatigue. The endplate noise is a consequence of the trigger point and is seen in the resting muscle.

Trigger points in Athletes: Reply: I added a comment that trigger points can be found in elite athletes.

“Muscle overload (muscle activity to or beyond fatigue) is a commonly accepted, though unproven, cause of TrPs. whether acute overload or chronic and repetitive. Thus, it would seem that elite athletes should all have TrPs. In fact, there is a literature related to treatment of athletes with TrPs; (see for example [62-65][Hidalgo 2013;Ortego-Santiago 2020; Ceballos-Laites 2021, Huang 2022].  However, there are no epidemiological studies of TrPs in athletes.”

Line 190: Reply: I agree with the reviewer, but this section I refer to the feedback mechanism sthat prevents a potential excess of calcium in the cytosol that overwhelms the reuptake mechanism and that can lead to muscle injury. They are discussed in more detail later in the paper. I do not see that a change in the text needs to be made here.

Line 219 and line 222: The reviewer is of course correct and I made the changes that were  recommended, as quoted in reply to section 2.1 above.

Tables 2 and 3. Reply: I agree with the reviewer and have removed those tables.

Line 241-242: Reply: I agree with the reviewer and have noted that in the text.

3.4 Re: PKC: I have amplified the comments about protein kinase second messenger systems so that they should be better understood by the reader.

3.7 Cantu syndrome. Reply: Thank you for suggesting that. I have added a brief description of the Cantu syndrome.

“Molecular subunit alleles in ion channels can not only up- or down-regulate channel function but can also alter sensitivity to inhibitory factors such as mutations in the KATP ion channel subunits in the Cantu Syndrome, a rare multisystem disorder with hypertrichosis, cardiomegaly, and skeletal and other anomalies [67]. “

3.7.2 Reply: The idea that there are mutations of both RyR and of KATP ion channels is an innovative idea that I do not know is correct or not, but one that I think is worth exploring because it may explain a vulnerablity to developing myofascial pain syndrome that is not now apparent.

4. Conclusion: Stress or muscle tension from incorrect postures: which of the 4 proposals would you enter? Reply: I am not quite sure what the reviewer means here, but I have rewritten the conclusion in keeping with the recommendation of the editor and have eliminated the specific points in favor of simply summarizing them in a general way.

Some words could be written differently:

Line 156: bupivicaine is better as bupivacaine

Ryanadine (lines 186 and 371) is better known as Ryanodine

Reply: Thank you. I have correct the spelling errors.

Reviewer 4 Report

It has been a real pleasure to read Dr. Gerwin's current update.

I am honored to be able to recommend his work for publication.

Author Response

I appreciate the reviewer's comment and am grateful for the support. I have revised the paper in accordance with other reviewer's recommendations to shorten the paper and to give it better focus. Thank you for taking the time to review this submission. 

Round 2

Reviewer 1 Report

Comments provided on the initial submission lacked specifics because the submitted manuscript was insufficiently ripe for review. To be clear this time, the entirety of the writing and presentation is unnecessary, weak, and/or diffusely focused. Thus, substantial revision in the manuscript presentation is needed before a proper review of the scientific merit can be conducted. Minimal changes were made to the initial submission; concerns described in the initial review persist.

While the author(s) are commended for attempting to synthesize the literature in support of Simons’ Integrated Hypothesis, there is much scientific concern undergirding the “New Hypothesis”. Multiple large and many small scientific errors throughout the manuscript collectively result in great doubt regarding the integrity of the ideas and overall scientific value. To cite just a few concerns, the present manuscript is problematic in its central theme due to what appears to be over-interpretation of the trigger point literature and a misunderstanding of muscle physiology. Additionally, the attention on mutations as a cause is simply bizarre given the ubiquity of trigger points.

An overarching belief presented in the manuscript is that trigger points consist of a segmental muscle contraction. Support for this theory comes from work by Simons and Stolov (1976) and Gerwin et al (2019) (1,2). While both of those studies present a beautiful histological micrographic image of segmental contraction taken from tissue at a trigger point, a reading of the papers shows the image to be the exception, not the rule. The data presented by both papers refute the notion of segmental contraction in trigger points. In Simons and Stolov, a greater total segmental contraction count difference as determine histologically was noted between control and test locations but the authors state “The larger total count in test biopsies is completely accounted for by one exceptional value.” The authors go on to conclude “None of the histological features measured in the study were convincingly related to palpable bands.” Gerwin et al report only 2 of 14 (14%) histological samples obtained from contraction nodules exhibited segmental contraction, a percentage that is unconvincing for the existence of segmental contraction. Furthermore, Gerwin et al note a higher percentage of control samples (4 of 14, 29%) exhibited segmental contraction. Given the importance of segmental contraction to the thesis of the “New Hypothesis” the base evidence to support this position is not justified.

A second over-arching belief in the manuscript is that trigger points are formed following eccentric muscle contraction. No empirical, subjective, or logical evidence for this exists. It appears that eccentric contraction is offered in the paper as a mechanism for trigger point induction because “the end justifies the means”, that is, eccentric contraction can result in segmental contraction and segmental contraction is a hypothesized key element of trigger points, therefore eccentric contraction must be an initiator of trigger points. However, eccentric contraction is a relatively rare contraction in typical human muscle activity, yet trigger points are common in skeletal muscle, including muscles that are unlikely to undergo eccentric contraction (temporalis, masseter). A far more likely initiator of trigger points is the long-held belief by researchers and clinicians that they form from concentric or isometric contraction elicited by repetitive use or “tension” (sustained muscle contraction). Office work and stress are common examples of trigger point induction and have been shown empirically to induce trigger points (3). While eccentric muscle contraction as a method of inducing trigger points should not be completely ruled out, stronger evidence is needed before it can be a main player in a theory such as presented here.

The manuscript stresses the role of the ryanodine receptor as critical for over-flooding the muscle cell with calcium ions, resulting in sustained contraction at the trigger point. This portion of the manuscript indicates a misunderstanding of muscle physiology and ryanodine receptors specifically. In cardiac and diaphragm muscle approximately 40% of the calcium involved with muscle contraction enters the cells through the ryanodine receptor (RyR2), while the remaining 60% originates from the sarcoplasmic reticulum. In skeletal muscle, however, the ryanodine receptor is RyR1 and has a trivial role with calcium influx; less than 5% of the calcium in the cytosol is attributable to the ryanodine receptor (5). Thus, the ryanodine receptor (RyR1) is a bit player in direct skeletal muscle calcium influx and errors in the ryanodine receptor are unlikely to be a significant factor in a trigger point. The notion that trigger points result from a “mutation” in the ryanodine receptor needs direct evidence.

Comments above identify just a few concerns with the “New Hypothesis”. A thorough review of the scientific merits has not been conducted.

(1)    Simons DG, Stolov WC. Microscopic features and transient contraction of palpable bands in canine muscle. Am J Phys Med. 1976;55(2):65-88.

(2)    Gerwin RD, Cagnie B, Petrovic M, Van Dorpe J, Calders P, De Meulemeester K. Foci of segmentally contracted sarcomeres in trapezius muscle biopsy specimens in myalgic and nonmyalgic human subjects: preliminary results. Pain Med 2020;21(10):2348-2356.

(3)    Treaster, D., W.S. Marras, D. Burr, J.E. Sheedy, D. Hart. Myofascial trigger point development from visual and postural stressors during computer work. Journal of Electromyography and Kinesiology 16 (2006) 115–124.

(4)    Balderas-Villalobos J., Steele T.W.E., Eltit J.M. (2021) Physiological and Pathological Relevance of Selective and Nonselective Ca2+ Channels in Skeletal and Cardiac Muscle. In: Zhou L. (eds) Ion Channels in Biophysics and Physiology. Advances in Experimental Medicine and Biology.

Author Response

Response to reviewer 1

I want to thank the reviewer for raising serious questions about the concept.

  1. I agree that the paper was overlong and poorly focused. I have rewritten the paper, shortened it by about 1/3, and focused the writing to be clearer. I have eliminated the distracting comments and repetitions.
  2. The reviewer questions the interpretation of the trigger point literature and my understanding of muscle physiology and the possible role of mutations/variants.
    1. Trigger points are indeed ubiquitous. I was careful to say that the proposed hypothesis may apply only to a subset of trigger points as there are likely multiple factors that could lead to a vulnerability to develop trigger points. [Introduction,, first paragraph]
    2. The reviewer suggests that I have overstated the role of segmental sarcomere compression [Section 2.4]. . To be sure, the literature is sparse in this regard. Simons and Stolov looked at taut bands so they were not focused on trigger points in their study. They looked at the taut bands of 10 dogs, and did not focus on trigger points.
    3. The study that I conducted was on material taken for another purpose (other morphological aspects of trigger points) and we had limited tissue available to examine. The sampling technique was also not ideal for being certain that trigger areas were biopsied. The reviewer notes that specimens with sarcomere contraction were more common in controls, but the controls all had taut bands that were biopsied, so that there was no contol muscle biopsied that was not at least a taut band. So segmental sarcomere contraction might be expectd in all tissue samples in this study. 
    4. To this date, there has not been a systematic examination of either animal or human material to answer the question about segmental sarcomere compression. However, there are several reports from animal models that report the same finding, and they are cited in the current version of the paper. Therefore, though not certain, there is some support for the concept of segmental sarcomere contraction. Finally, segmental sarcomere contraction has been reported with subthreshold acetylcholine release, both is cardiac and skeletal muscle, references again cited in the paper, so the finding is certainly consistent with what has been reported, but I agree with the reviewer that the finding still needs to be confirmed.
    5. There are two major points that I make in the paper with regard to muscle physiology. (Section 3.6  and in section 3.7] They are relevant to the long held notion among clinicians that muscle overload or exercise beyond the tolerance of the muscle to work can lead to trigger points. I cite literature that shows that intense muscle exercise can indeed be injurious to muscle. I also cite literature that supports the concept that there are mechanisms that designed to limit the possibility of muscle damage by limiting the frequency of evoked muscle action potentials when ATP levels are low, and mechanisms that inhibit the further infux of calcium into the muscle cell when ATP levels are low. The first is the interaction between adenosine and muscarinic receptors and the second is mediated by the KATP ion channel. These are described in detail in the paper. The hypothesis suggests that these systems are inadequate to prevent the development of trigger points. I suggest that one possible reason for their failure to protect muscle is that one or more such systems is not functioning properly, because of mutation/variant in one of the components of an involved receptor, or because of a post-translational functional change.
    6. The reviewer suggests that the attention on mutations is bizarre given the ubiquity of trigger points. I agree with the reviewer that trigger points are ubiquitous. However, I do not claim that the Hypothesis applies to all trigger points. There are reports of exertional myalgia and exercise intolerance and decreased exercise capacity in both mice and in humans associated with dysfunctional RyR1 channels, either because of mutations or because of post-translational modification of the channel. I cite these and comment on their relevance, particularly with regard to exercise induced trigger points. [ Sectio 3.7.2 and 3.7.3] In none of the cited studies were the animal or human subjects examined for trigger spots or trigger points.  Also, KATP knockout mice show unusual fatigue to intense exercise and also show segmental sarcomere contraction, making the possiblity of coexistant trigger points intriguing but unanswered.
    7. The reviewer says that I believe that eccentric lengthening of muscle is a major cause of trigger points. I mentioned eccentric lengthening of muscle as an example of muscle overuse, not as a major cause of trigger points in general. [Section 3.1] Persons who hike long distances downhill or who water ski are subject to the stress of eccentric lengthing of the quadriceps and are known to have muscle pain after such activity. I cite studies that confirm that muscle damage can occur after eccentric lengthening, indicating that it can certainly be stressful to muscle. However, reports of trigger points in such subjects is incidental at best, and I know of no actual study of this problem. I make clear in the revision that eccentric lengthening is to be taken as an example of muscle overload.
    8. The reviewer says that I misunderstand the role of the RyR1 receptor in muscle contraction. [Section 3.2 and elsewhere] The reviewer cites studies on cardiac and diaphragm muscle to make the point that the RyR1 channel is of little importance in Ca2+ influx into myofibril cytosol. However, the ryanodine receptor isoforms are differentially distributed in tissues and their activation and modulation is different in different tissues. Whereas the RyR1 isoform is predominant in most skeletal muscle, the RyR2 isoform is predominant in  cardiac muscle. In both skeletal muscle and cardiac muscle the RyR channel is noted to be the major channel through which Ca2+ enters the cytosol from the SR and initiates contraction. In cardiac muscle, the process is CICR (calcium induced calcium release), which plays a minimal role in skeletal muscle. I could not find a reference which said that RyR1 channel ativity plays a minor role in the ECC process, as the reviewer suggsts (the reviewer’s reference 5 which was not provided by the reviewer). I have quoted (perhaps at too great length) 5 artricles that each say that the RyR1 channel is the main channel for the influx of Ca2+ needed for ECC. And these are only a sample.
    9. There is literature to support the role of the RyR1 ion channel in excitation-contraction coupling. I cite below 5 examples of articles which state that the membrane depolarization activates the DHPR receptor that then activates the RyR1 receptor that when open allows Ca2+ influx into the cytosol from the SR that is necessary for muscle contraction. That is all that I am trying to say in the article. I also realize that Ca2+ is also stored and released from the Endoplasmic Reticulum and from mitochondria, but they are not central to the theme of this article.
      1. Balderas-Villalobus et al. as cited by the reviewer states “When RyR1 opens, the large chemical gradient drives Ca2+ efflux from the SR into the cytosol to activate the nearby contractile machinary.” The article further says that “RyR1 is an important indirect regulator of the Ca2+ permeability in the plasmalemma.” as an example of the role of a non-channel transit of Ca2+ into the cytosol. Regarding dysfunctional (mutant or otherwise altered channels), the authors say, “ Ca2+ enter at rest is enhanced in knock-in mice models expressing RyR1 mutations linked to MH [malignant hyperthermia] syndrome. Leaky RyR1s may decrease SR Ca2+ stores, thereby increasing the plasma membrane permeability to Ca2+. This results in a chronically elevated [Ca2+]cyt at rest. The chronic elevation of Ca2+ produces oxidative stress resulting in a metabolic toll on the muscle fiber. The presence of a leaky RyR1…may also promote a compensatory effect in the overexpression of nonselective TRPCs, thereby further contributing to the misregulation of resting Ca2+.”
      2. Ballinger et al. Proc Natl Acad Sci USA 2008; 105:2198-2022, states that “during exercise in mice and humans the major Ca2+ release channel required for excitation-contraction, the ryanodine receptor (RyR1)…”
      3. Dulhunty AF et al. Clinical & Experimental Pharmacology & Physiology, 2017;44(1):3-12. “The core skeletal muscle ryanodine receptor (RyR1) calcium release complex extends through three compartments of the muscle fibre, linking the extracellular environment through the cytoplasmic junctional gap to the lumen of the internal sarcoplasmic reticulum (SR) calcium store. The protein complex is essential for skeletal excitation‐contraction (EC)‐coupling and skeletal muscle function.”

                 “ Nevertheless it is irrefutably established that coupling between depolarisation and      “Ca2+ release depends on expression of the skeletal isoforms of the DHPR α1S and β1a and RyR1.[ 1] , [ 2] , [ 10]”

4. Sorrentino V. Sarcoplasmic reticulum: structural determinants and protein dynamics. Intl J Biochem Cell Biol 2011;43:1075.  “Following the depolarization of the plasmamembrane, activation of DHPRs on the T tubule triggers the opening of the RyRs on the SR. Activation of RyRs releases massive amounts of Ca2+ from the SR that start muscle contraction, hence the entire process is called excitation–contraction (E–C) coupling (Rios et al., 1991, Lanner et al., 2010).”

5.   Woll and van Petergem. Physiologic reviews 2022;102(1):209-268. “Ca2+ is a vital ion involved in numerous intracellular signaling processes throughout the human body. From releasing neurotransmitter and hormones, to mediating muscle contraction, shaping action potentials, and regulating gluconeogenesis and migration, this tiny ion has proven to be a potent second messenger in a wide array of processes. In fact, it is hard to find a signaling pathway that is not influenced by Ca2+ ions, directly or indirectly. Their importance can also be understood by the effort of each cell to keep the cytosolic free Ca2+ concentrations very low, at 100 nM and below under resting conditions. This is in great contrast with the millimolar concentrations found in the extracellular milieu. A temporary opening of Ca2+-permeable channels can therefore rapidly increase the cytosolic free Ca2+ concentration by more than an order of magnitude, into the micromolar range, triggering signals involving proteins that are tuned to this concentration range. In addition to this, several organelles form intracellular Ca2+ stores, for which the endoplasmic reticulum (ER), as well as the sarcoplasmic reticulum (SR) in muscle cells form large pools, storing Ca2+ at up to millimolar concentrations (1). Release of Ca2+ from these stores, often in conjunction with extracellular Ca2+ entry, has become a cornerstone in many physiological and pathophysiological signaling processes.

The membranes of the ER and SR are home to the inositol-1,4,5-trisphosphate receptor (IP3R), and the ryanodine receptor (RyR), both known as Ca2+-release channels.” “A unique trigger also exists for RyR1, the predominant RyR isoform expressed in skeletal muscle. Although it can be stimulated to open via cytosolic Ca2+, it also is able to receive input from another membrane protein, located in the plasma membrane. Depolarization of the plasma membrane results in conformational changes in CaV1.1, the L-type voltage-gated calcium channel in skeletal muscle. This leads to opening of RyR1 via mechanical interactions, without the need for initial Ca2+ influx from the extracellular space (27). It requires several additional proteins, which together form a large complex, consisting of four CaV1.1 channels per homotetrameric RyR1 (28). The process of communication between L-type voltage-gated calcium channels and RyRs in muscle tissue is also known as excitation-contraction (EC) coupling, as it is the key step in converting an electrical signal (depolarization of the plasma membrane) into a chemical one (a rise in cytosolic Ca2+).”

5. Murayama T and Ogawa Y. Trends in Cardiovascular Medicine 2002;12(7):305-311. “Depolarization in the sarcolemma causes a conformational change of DHPR, which is transmitted to RyR1 to trigger Ca2+ release from RyR1 (depolarization-induced Ca2+ release; DICR) (Schneider 1994). This mode of Ca2+ release is of physiological relevance in muscle contraction by electrical stimulation. In this case, DHPR works as a voltage sensor…”

7. The reviewer says that the notion that trigger points result from mutations [or modifications of the RyR1 receptor needs direct evidence. I absolutely agree with that. I provide evidence in Sections 3.7.2 and 3.7.3, in humans and in rodents.  I am simply suggesting that this is a possibiltiy for some trigger points, and would hope that this proposal might lead to someone actually studying this. I happen to be retired and do not have the ability to study this directly myself, but hope to stimulate interest in the possibility so that it can be examined, and either refuted or confirmed. I therefore offer the idea as a means to get people thinking about the possibility. I may be wrong, but there is enough of a suggestion in the literature to think that some of these mechanisms may be operative.

I hope that the revised version of the manuscript answers some of the concerns this reviewer has. I thank the reviewer for helping me focus the paper and to address some of the central issues more clearly.